# Advances in Genetic Transformation of Lactic Acid Bacteria: Overcoming Barriers and Enhancing Plasmid Tools

**DOI:** 10.3390/ijms26189146

**Published:** 2025-09-19

**Authors:** Aleksei S. Rozanov, Leonid A. Shaposhnikov, Kseniya D. Bondarenko, Alexey E. Sazonov

**Affiliations:** Scientific Center of Genetics and Life Sciences, Sirius University of Science and Technology, Sirius 354340, Russia; bondarenko.kd@talantiuspeh.ru (K.D.B.); sazonov.ae@talantiuspeh.ru (A.E.S.)

**Keywords:** lactic acid bacteria, genetic transformation, electroporation, natural competence, plasmid vectors, restriction–modification systems, conjugation, shuttle vectors

## Abstract

Lactic acid bacteria (LAB) are central to food fermentation, probiotic delivery, and emerging synthetic biology applications, yet their robust cell envelopes and restriction–modification systems complicate DNA uptake. This review synthesizes practical routes for introducing DNA into LAB—natural competence, electroporation, conjugation, phage-mediated transduction, and biolistics—and outlines vector systems for expression and chromosomal editing, including food-grade strategies. We highlight recent advances that broaden strain tractability while noting strain-to-strain variability and host-specific barriers that still require tailored solutions. These advances directly enable applications in food and probiotic biotechnology, including improving starter robustness, tailoring flavor and texture pathways, and installing food-grade traits without residual selection markers. We close with near-term priorities for standardizing protocols, widening replicon compatibility, and leveraging modern genome-editing platforms to accelerate safe, marker-free engineering of industrial and probiotic LAB.

## 1. Introduction

*Lactic acid bacteria* (LAB) are a diverse group of Gram-positive microbes extensively used in dairy fermentation, food preservation, and as probiotics. Their fermentative metabolism contributes to yogurt, cheese, and other fermented products, while some strains confer health benefits in the gut. Beyond gastrointestinal benefits, recent clinical and mechanistic syntheses suggest that selected probiotic LAB can modulate symptoms across several mental disorders via the microbiota–gut–brain axis (“psychobiotics”) [1]. Genetic engineering of LAB holds great promise for improving fermentation properties, enhancing nutritional or probiotic functions, and producing high-value metabolites. However, LAB are very difficult to manipulate genetically.

One factor underlying the notoriously low transformation efficiency of lactic acid bacteria is their cell wall architecture, particularly the thick peptidoglycan layer and associated polysaccharides. This robust cell wall acts as a physical barrier to exogenous DNA, severely impeding the uptake of foreign genetic material. As a consequence, genetic manipulation of LAB and molecular studies on these strains are significantly constrained by these structural defenses. Furthermore, many *Lactobacillus* strains exhibit high resistance to cell-wall hydrolases such as lysozyme and mutanolysin [2], complicating experimental approaches that rely on enzymatic cell wall digestion to facilitate DNA entry.

Another critical consideration in engineering LAB is their potent defense against foreign DNA, chiefly the presence of multiple restriction–modification (R-M) systems [3]. These R-M systems function as a natural immune barrier, recognizing and cleaving DNA that lacks the host’s specific methylation patterns. In practice, this means incoming plasmids propagated in a typical *E. coli* host can be rapidly degraded unless their DNA methylation profile is compatible with the LAB’s R-M enzymes. Consequently, to obtain LAB transformants, researchers often must use plasmid DNA prepared in *E. coli* strains lacking key DNA methylation (Dam/Dcm) activities or in *E. coli* strains whose methylation pattern matches that of the target LAB strain [3]. Such strategies help the plasmid DNA evade host restriction; indeed, using unmethylated plasmid DNA (from a Dam^−^/Dcm^−^ *E. coli* strain) has been shown to boost transformation efficiency in LAB by over three orders of magnitude.

Given these obstacles, two general approaches are employed: exploiting any native competence pathways and using artificial DNA delivery methods. Historically, most LAB (especially *Lactobacillus*, *Bifidobacterium*, etc.) were not known to be naturally competent, meaning they could not readily take up free DNA from their environment. However, landmark studies in the 2000s demonstrated that competence can be induced in some LAB under engineered conditions. For LAB species lacking a workable competence system, artificial transformation remains the primary solution. Electroporation is the simplest and most widely used method. Optimized electroporation protocols now exist for many LAB (including *L. lactis*, *L. casei*, *L. plantarum*, *L. brevis*, and *S. thermophilus*), achieving transformation efficiencies on the order of 10^4^–10^6^ colony-forming units per microgram of DNA. Such levels are sufficient for routine introduction of plasmids and recombineering constructs. Nevertheless, some wild or industrial LAB strains remain highly recalcitrant even to electroporation, due to extremely impermeable cell walls or aggressive R–M systems. When natural competence is unavailable, alternative delivery routes are used. Examples include optimized electroporation, conjugation, and phage-mediated transfer.

Beyond DNA delivery, reliable plasmid vectors are essential tools for LAB engineering. Researchers have developed shuttle vectors that carry origins of replication functional in both *E. coli* and various Gram-positive hosts, enabling easy cloning in *E. coli* and stable maintenance in LAB. Specialized plasmids have also been created for chromosomal integration and plasmid curing: for instance, temperature-sensitive (Ts) or helper-dependent systems can promote integration of “suicide” plasmids into the genome, allowing gene knockouts or insertions without leaving foreign DNA behind. Overall, advances in transformation techniques and vector design have significantly improved our ability to manipulate LAB genomes, though many species still require tailored strategies.

Because LAB are used both industrially and as promising live biotherapeutics, better DNA delivery and vector systems have immediate applied consequences: (i) more resilient starter cultures (e.g., phage tolerance, stress robustness), (ii) rational flavor/texture optimization via metabolic rewiring, and (iii) probiotic chassis capable of food-grade antigen/enzyme delivery. Section 2, Section 3, Section 4, Section 5 and Section 6 synthesize routes for DNA entry (natural competence, electroporation, conjugation, phage-mediated transduction, and biolistics), while Section 7, Section 8 and Section 9 summarize plasmid toolkits for expression and chromosomal editing. We also briefly signpost emerging genome-editing platforms in LAB and refer readers to a dedicated review for an in-depth treatment [4].

## 2. Natural Transformation and Competence

Natural transformation is a process by which bacteria actively take up free DNA from their environment and incorporate it into their genomes. This process requires a physiological state called competence, wherein a set of genes for DNA uptake and integration are expressed. In competent cells, double-stranded DNA binds to the cell surface, and one strand is translocated into the cytoplasm, while the other strand is degraded. The incoming single-stranded DNA is immediately coated by RecA and DprA proteins and then integrated into the chromosome by homologous recombination [5,6,7]. Natural competence is widespread in some bacterial groups—notably, *Streptococcus* and *Bacillus* species—and has been exploited for genetic engineering for decades. *Streptococcus pneumoniae*, though a pathogenic streptococcus, is a classic model: it has been routinely manipulated by natural transformation for over 60 years [7]. As a historical note, pneumococcal transformation traces back to Griffith’s experiments [8], which set the stage for DNA’s role in heredity later demonstrated by Avery, MacLeod and McCarty [9]; for contemporary food microbiology overviews, see [10,11].

The molecular mechanisms underlying natural competence in lactic acid bacteria involve a sophisticated regulatory cascade initiated by quorum-sensing-mediated signal detection. Competence pheromones, typically small peptide molecules, bind to cognate membrane-bound receptor complexes, predominantly histidine kinases of two-component regulatory systems, triggering autophosphorylation and subsequent phosphotransfer to their cognate response regulators [12]. This phosphorylation event activates transcription of the master regulatory gene *comX*, encoding an alternative sigma factor that orchestrates the competence regulon [13,14]. Upon accumulation, ComX associates with the RNA polymerase core enzyme to form a holoenzyme that specifically recognizes combox sequences within promoter regions of competence-associated genes [15]. This transcriptional reprogramming results in coordinated expression of multiple gene clusters essential for DNA uptake machinery, including membrane transporters (ComEA, ComEC) and processing endonucleases (*nucA*, *endA*), recombination apparatus (RecA, SsbB, RadA, DprA), and DNA-processing enzymes such as helicases, ligases, and polymerases [16]. The culmination of this regulatory cascade transforms the bacterial cell into a competent state capable of binding exogenous DNA at the cell surface, facilitating its transport across the cell envelope, protecting incoming DNA from nucleolytic degradation, and ultimately integrating foreign genetic material into the chromosome via homologous recombination pathways [17]. This process is schematically shown in Figure 1.

Many lactic acid bacteria, especially those used in dairy and fermentation, historically did not exhibit robust natural competence under laboratory conditions. This presented a barrier to genetic engineering, as one could not simply add DNA to a culture and obtain recombinants, unlike the ease of transformation in *B. subtilis* or *S. pneumoniae*. However, research in the mid-2000s demonstrated that some LAB can be coaxed into competence. A landmark example was *Streptococcus thermophilus*, a yogurt starter culture that was long considered non-competent. In 2006, Blomqvist and colleagues showed that *S. thermophilus* can be transformed at high efficiency by mimicking its natural competence system [7]. They added synthetic competence-stimulating peptide and overexpressed the ComX regulator, which together induced the competence genes in this species. As a result, *S. thermophilus* became capable of taking up plasmid DNA and even linear DNA fragments, enabling straightforward genomic edits [7]. It is noteworthy that DNA fragments up to ~15 kb have been integrated into *S. thermophilus* with ~1 kb homology arms—sizes typically impractical for direct plasmid cloning [18].

Other streptococci relevant to dairy and health have also been found to exhibit natural competence. For instance, certain strains of *Streptococcus thermophilus* vary in their transformability—some achieve very high frequencies of recombinants, whereas others show only ~1% of cells taking up DNA [18]. This strain-dependence suggests that minor genetic differences (perhaps in regulatory circuits or membrane proteins) can affect competence; ongoing research is exploring how to improve low performers. Meanwhile, researchers extended the competence induction strategy to *Lactococcus lactis*, a species that is not naturally competent in the wild. By overproducing the ComX sigma factor in *L. lactis*, David B. and colleagues demonstrated that *Lactococcus lactis* subsp. *cremoris* KW2 can be made artificially competent [13]. In their study, *L. lactis* subsp. *cremoris* KW2 strain was engineered to express a homolog of ComX, which in turn activated expression of DNA uptake machinery, resulting in detectable transformation with exogenous DNA.

Despite these advances, natural transformation is currently feasible in only a few LAB (certain *Streptococcus* and *Lactococcus*). For the majority of LAB species (e.g., most *Lactobacillus, Bifidobacterium,* etc., which lack known competence systems), artificial DNA delivery remains the method of choice.

The advent of natural competence in a few LAB species has represented an important advance for those particular cases. *S. thermophilus* now joins *B. subtilis* as a Gram-positive organism in which genome editing can be done by direct uptake of PCR products, greatly accelerating strain development [7]. This is especially valuable for dairy industry strains, as it permits rapid, marker-free genetic improvements (for example, deleting phage receptor genes to make starters phage-resistant, or knocking out metabolic regulators to enhance flavor compound production) without leaving antibiotic resistances. Moreover, naturally competent LAB can potentially acquire large genomic islands or pathways in one step, facilitating metabolic engineering. The ability to construct food-grade mutants via natural transformation—as highlighted by Blomqvist—benefits consumer safety and regulatory acceptance [7]. On the other hand, the limitation is that true natural competence is still the exception rather than the rule among LAB. Unlike *S. pneumoniae* or *B. subtilis*, which become naturally competent under certain conditions, most LAB require significant genetic tweaking (e.g., introducing ComX) or are simply not transformable. Even in *S. thermophilus*, competence induction may require strain-specific fine-tuning (peptide concentration, growth phase, etc.), and not all strains respond equally [18]. For those LAB where natural transformation is not (yet) achievable, electroporation remains an effective, if less “elegant,” solution.

## 3. Electroporation

Electroporation is rapid and broadly applicable. A short electrical pulse creates transient pores in the cell envelope, allowing exogenous DNA to enter. However, beyond the electrical pulse itself, it is important that the rigid cell wall of Gram-positive bacteria be pre-weakened using chemical agents added during culture growth. Other critical factors include the washing buffer, recovery buffer, composition of media, cultivation conditions, electric pulse parameters, and the growth stage and density of the cells. Unfortunately, for LAB many of these parameters vary from strain to strain. Electroporation remains the most widely used artificial method in LAB; see [19] for a consolidated review of parameters and troubleshooting across species. Schematically electroporation is shown on Figure 2.

When establishing electroporation protocols, one should first determine the growth rate of the strains and identify the optimal growth phase for harvesting cells. Typically, cells from either the exponential or stationary phase are used for making competent cells [20,21]. For example, *Lactobacillus delbrueckii* subsp. *bulgaricus* VI104 harvested in early stationary phase (OD_600_ ≈ 1.7) yielded about 750 transformants/µg DNA—3.75-fold higher efficiency than cells from the exponential phase (OD_600_ 0.2–1.0). Moreover, higher cell density can markedly improve yields: at 10^10^ CFU/mL, the number of transformants was ~50× greater than at 10^9^ CFU/mL and ~385× greater than at 10^8^ CFU/mL (illustrating the benefit of using a high concentration of competent cells).

Common lactic acid bacteria are cultured on a few rich media optimized for their nutritional needs. De Man, Rogosa and Sharpe (MRS) medium is the standard for cultivating *Lactobacillus* spp., providing carbon (glucose), nitrogen, vitamins, and minerals ideal for fastidious *lactobacilli*. Many *Lactobacillus* strains (e.g., *L. plantarum*) grow robustly in MRS, which supports their complex nutritional requirements. *Bifidobacterium* spp. will also grow on MRS-based media, but typically with added L-cysteine (0.05% *w*/*v*) to lower redox potential—reflecting their anaerobic nature [22]. In contrast, *Lactococcus* and *Streptococcus* are often grown on M17 medium (originally formulated for lactic streptococci); when supplemented with 0.5% glucose (termed GM17), it efficiently supports *lactococci* like *L. lactis* and streptococci such as *S. thermophilus*. For example, *L. lactis* is routinely propagated in GM17, while *S. thermophilus* may be grown in M17 with lactose as the sugar (sometimes called LM17). Other rich broths like Brain Heart Infusion (BHI) are also used for some streptococci, but MRS and M17/GM17 remain the most widely used media for LAB. Each medium’s formulation suits a different LAB group: MRS favors acid-tolerant, nutritionally demanding *lactobacilli* (and *bifidobacteria* with modifications), whereas M17/GM17—which contains peptides, yeast extract, and buffering glycerophosphate—is optimal for the metabolic profiles of *Lactococcus* and related genera. These media thus represent “optimal” laboratory diets for their respective LAB members, ensuring reliable growth and high viability in research and industry cultures.

Another important step is choosing chemical agents to weaken the cell wall and make cells more permeable prior to electroporation. Common cell wall-weakening agents include:

**Glycine**: One of the most popular choices for Gram-positive bacteria. Glycine can substitute for L-alanine and D-alanine in peptidoglycan precursors, inhibiting cell wall crosslinking and causing cells to grow with morphological abnormalities. The resulting cell wall (lacking the methyl group present in alanine) is looser and more porous, which favors uptake of macromolecules. Adding glycine to the culture medium can dramatically enhance electrotransformation efficiency, though the optimal concentration must be determined for each strain. For example, *L. plantarum* NCFB1193, NCFB343, and FMT600 showed a 30–100-fold increase in transformant numbers when grown in MRS with 6% glycine to mid-exponential phase [23]. Many *L. casei* strains also exhibit high transformation efficiencies when grown with 0.5–1% glycine [21]. In general, low glycine levels may slightly inhibit growth, but increasing the glycine concentration tends to improve competency. For *L. plantarum* NCFB1193, the electroporation efficiency rose significantly as glycine in the growth medium was raised from 4% up to 14%, with ~8% being optimal in that study. When using higher glycine levels, an osmoprotectant—typically 0.5 M sucrose—should also be included in the wash and recovery media to prevent lysis of the glycine-weakened, swollen cells and to maximize post-pulse survival [21,24].

**Sodium chloride (NaCl)**: High salt can also increase cell wall permeability. NaCl has been shown to reduce the content of lipoteichoic acids, shorten peptidoglycan chain length, and decrease D-alanine substitutions in the cell wall, thereby disrupting the cell’s normally ordered envelope structure. A high-osmolarity environment weakens and even partially lyses the cell wall, which facilitates pore formation during electroporation. Salt also imposes osmotic stress that can cause the cell wall to rupture, releasing cellular contents. Effective NaCl concentrations vary by strain, but commonly 0.7 M–0.9 M NaCl is used. Palomino and colleagues developed a high-salt method for *Lactobacillus casei*: growing cells in MRS containing 0.9 M NaCl produced “salt-stressed” *lactobacilli* with more permeable walls. As a result, electrocompetent *L. casei* yielded up to ~10^5^ transformants/µg DNA—a substantial improvement over traditional protocols [25]. This method proved effective across several *Lactobacillus* species (transforming *L. paracasei, L. plantarum, L. acidophilus*, etc. at similar efficiencies). Later, Welker and colleagues obtained comparable benefits in *Lactobacillus*: including 0.9 M NaCl during growth significantly boosted electrotransformation efficiency in *L. casei* strains, on par with the improvements seen using cell-wall inhibitors like glycine [21]. In Welker’s study, salt-grown cultures showed markedly higher colony counts after electroporation than unsupplemented controls. These examples demonstrate that NaCl-induced osmotic stress can transiently “soften” the rigid Gram-positive cell envelope, facilitating plasmid entry.

**Lysozyme**: Lysozyme is an enzyme that hydrolyzes specific β-1,4-glycosidic bonds in peptidoglycan (note that peptidoglycan has two types of β-1,4 linkages: between GlcNAc C-1—MurNAc C-4, and between MurNAc C-1—GlcNAc C-4; lysozyme cleaves only the latter bond). Thus, lysozyme can fragment the peptidoglycan into GlcNAc and MurNAc subunits once cells have grown to a certain stage, thereby weakening mature cell walls. Treating cells with lysozyme before electroporation can dramatically improve efficiency. For instance, *Streptococcus lactis* LM0230 treated with 2000 U/mL lysozyme at 37 °C for 20 min had 30–1000× higher transformation efficiency than untreated cells [24,26].

**Penicillin**: Penicillin is a structural analog of the D-alanyl-D-alanine dipeptide at the end of peptidoglycan pentapeptides. It irreversibly binds the transpeptidase (penicillin-binding proteins), blocking peptidoglycan crosslinking and causing defects in cell wall synthesis. This weakens the cell wall barrier. For example, treating cultures with 10 µg/mL penicillin led to a ~480-fold increase in electrotransformation efficiency [27,28]. However, penicillin-treated cells become enlarged and structurally fragile, so they should be handled gently.

Aside from glycine and lysozyme described above, some protocols have also used DL-threonine (at 20–40 mM) as a cell wall inhibitor; threonine can also induce cell wall stress responses that facilitate DNA uptake. van der Lelie first showed that incorporating 40 mM DL-threonine in the growth medium of *Streptococcus cremoris* generated spheroplast-like cells amenable to electroporation [29]. In their work, threonine-treated streptococci yielded transformants whereas untreated cells did not, proving the cell-wall inhibition was key. Building on this, Rodriguez and colleagues optimized electroporation of *Pediococcus acidilactici* by growing the cells with 20–40 mM DL-threonine. They observed that raising the threonine from 20 to 40 mM roughly doubled the transformation efficiency of *P. acidilactici* (from ~1 × 10^4^ to ~2 × 10^4^ CFU/µg in their system) [30]. Similarly, in *Clostridium* (another Gram-positive genus), adding 20–40 mM DL-threonine during growth provided ~1.6- to 2.1-fold increases in electrotransformation efficiency (albeit higher threonine had diminishing returns) [31]. The likely mechanism is that excessive threonine disrupts assembly of cell-wall precursors, thinning the peptidoglycan layer. However, there are trade-offs: scientists noted that while 40 mM threonine enabled various Gram-positive bacteria to be transformed, the absolute frequencies remained modest, and growth may be somewhat inhibited. In practice, DL-threonine (20–40 mM) has been used as an optional pre-treatment to reproducibly improve DNA uptake in various LAB—from *Lactococcus* and *Lactobacillus* species to others—by generating “leaky” cells more receptive to foreign DNA.

Choosing an optimal washing buffer for competent cells is critical. The most popular option for many LAB is a sucrose solution, which acts as an osmotic stabilizer. Sucrose prevents cells from taking up excessive water and bursting due to the hypotonic shock of electroporation, thereby improving cell survival [21,24]. For instance, Holo and Nes’s classic *L. lactis* protocol used 0.5 M sucrose as the washing buffer. Some protocols use a combination of sucrose and glycerol to ensure proper osmotic pressure and membrane protection. In fact, at very high cell densities and high field strengths, including ~10% glycerol in the wash buffer can prolong the discharge time of the electric pulse and effectively reduce arcing during electroporation [23]. It is also recommended to add MgCl_2_ (e.g., 2–10 mM) to washing or electroporation buffers—magnesium ions help remove or neutralize extracellular polysaccharides that might otherwise bind DNA on the cell surface, thereby increasing the amount of DNA actually entering the cells.

Not only is sucrose useful in the wash, but it can also be included in the recovery medium after electroporation. For example, adding 0.5 M sucrose to the recovery broth helps osmotically stabilize cells that were weakened by glycine treatment and the electric shock, improving post-pulse survival. By analogy with yeast transformation protocols, some LAB protocols also incorporate a brief treatment with lithium acetate (LiAc) and dithiothreitol (DTT); this combination (commonly used to make yeast spheroplasts more permeable) can transiently increase the porosity of LAB cell walls. Indeed, pretreatment with LiAc/DTT has been successfully used to boost electroporation efficiencies in *L. lactis*, *L. plantarum*, and *L. buchneri* mimicking the yeast protocol [32,33].

Recovery medium and incubation time are critical. A 2–3 h outgrowth in rich, non-selective medium typically balances repair with plasmid retention. It has been reported that using a special *Streptococcus* Recovery (SR) medium significantly improved the post-electroporation outgrowth and regeneration of *L. lactis* compared to standard media like GM17 + 0.5 M sucrose (SGM17) or GM17 alone. For instance, *L. lactis* BC101 yielded ~3× more transformants on SR recovery medium than on glucose-M17 (GM17) [34].

Equally important is the recovery incubation duration. Generally, after electroporation, cells are incubated in a rich, non-selective recovery medium for 1–3 h before plating on selective agar. During this time, cells repair membrane damage and begin to express antibiotic resistance or other markers from the introduced plasmid. Studies show that even 2 h can make a big difference: about one-third of *L. lactis* BC101 transformants became capable of growing on a hypotonic (no-sucrose) medium after 2 h of recovery [34]. The increased survival and colony formation in this period are likely due to both repair of electroporation-induced damage and initial rounds of cell division. However, the recovery time should not be extended for too long. In the recovery broth (which lacks selective pressure), both plasmid-bearing and plasmid-free cells will multiply. Typically, plasmid-free LAB grow faster than those carrying a plasmid, and even transformed cells can lose the plasmid if they undergo many generations without selection. Therefore, prolonged recovery can reduce the fraction of cells that maintain the plasmid, ultimately lowering the number of colonies on selective plates. In practice, 2–3 h recovery is a good compromise to allow good growth while minimizing plasmid loss.

Key electroporation conditions include the field strength, pulse duration, resistance, and current. Optimal values can differ depending on growth phase. In one study, the highest transformation efficiency for *Lactococcus lactis* (strains BC101 and IL1403) was achieved at an electric field intensity of ~12.5 kV/cm; for cells in mid-exponential phase the peak efficiency shifted to ~13 kV/cm, whereas late-exponential or stationary-phase cells required ~15 kV/cm for best results. Similarly, using a higher circuit resistance can sometimes improve efficiency: for several *Lactobacillus casei* strains, a 400 Ω resistance yielded higher transformation rates than 200 Ω under the same voltage conditions [34]. These examples underscore that fine-tuning pulse parameters (field strength and resistance in particular) is necessary for each strain and growth condition. Table 1 below summarizes and compares different factors and how they affect electroporation in LAB.

In summary, when planning electroporation experiments for LAB, one must consider a range of factors: the growth rate of the culture, the cell density and growth phase at which cells are made competent, the choice of cell wall-weakening agents and their concentrations, the composition of growth and recovery media, and the buffer used for washing cells. Special attention should be given to the electroporation parameters themselves (voltage, resistance, pulse length, etc.). Finally, one should remember that each LAB strain is unique and may require individualized optimization of every step—from competent cell preparation to post-electroporation recovery—in order to achieve successful transformation.

## 4. Conjugation of LAB

Conjugation in LAB refers to the direct transfer of DNA (usually plasmids or integrative elements) through cell contact. Many LAB harbor conjugative plasmids or integrative and conjugative elements (ICEs) that encode the machinery for their own transfer (type IV secretion systems, oriT sites, etc.) [36]. These mobile genetic elements often carry traits important for dairy fermentation—for example, in *Lactococcus lactis*, many genes for lactose metabolism, proteinases, and exopolysaccharide production reside on conjugative plasmids. Likewise, certain bacteriocin production (e.g., nisin biosynthesis) and sugar utilization operons are carried on ICEs. Mobilizing such elements by conjugation can combine beneficial traits into a single strain or alter a strain’s phenotype, underscoring the practical value of conjugation-based methods in LAB improvement.

In conjugation, DNA (usually a plasmid) is transferred directly from a donor to a recipient cell through physical contact. In the canonical F-plasmid system, the donor cell expresses *tra* genes that build a Type IV secretion system (T4SS) and a conjugative pilus [37]. A relaxase enzyme (TraI) nicks one strand of the donor’s plasmid at the origin of transfer (oriT) and pilots that single-stranded DNA (the “T-strand”) through the pilus into the recipient. Meanwhile, the plasmid in the donor undergoes rolling-circle replication so that a new complementary strand replaces the transferred one. Once inside the recipient, the incoming single-stranded DNA is coated and converted back to double-stranded form by host enzymes [37,38]. This process is schematically shown in Figure 3.

Conjugation efficiency also depends on several biological and experimental factors. For example, donors and recipients are typically grown to mid-log (exponential) phase and mixed at a defined ratio (often ~1:1) to maximize contact [38]. Other important conditions include cell density, medium composition, temperature, and incubation time. Mating is usually done at a temperature that supports growth (often ~30–37 °C for common LAB) and can take several hours. Studies show that conjugation frequency drops dramatically in stationary phase or at suboptimal temperatures, whereas log-phase cells under optimal conditions (e.g., 37 °C, rich medium) yield the highest transfer rates. In practice, donors and recipients are often concentrated (e.g., by centrifugation) before mating to increase encounter rates [38].

Conjugation assays in LAB are typically performed by co-incubating donor and recipient cells on solid or liquid media. Common methods include:**Filter mating**: A mixed suspension of donor and recipient cells is spotted onto a sterile membrane filter (e.g., 0.22 µm nitrocellulose) on an agar plate. After incubating (often 6–24 h at ~37 °C), the cells are washed off the filter and plated on selective media. This method concentrates cells in a tight spot to promote contact [38,39]. For example, one protocol mixes donor and recipient cultures, pellets them, resuspends in a small volume, and places the drop on a filter on pre-warmed agar; after ~6 h at 37 °C, cells are recovered for plating [39].**Agar (plate) mating**: Donor and recipient are mixed and spread directly onto a nutrient agar surface without a separate filter. The solid surface keeps cells in close proximity as they grow, allowing conjugation [38]. After incubation (e.g., overnight), bacteria are scraped or washed off and plated on selective media. This method is essentially similar to filter mating but omits the membrane.**Liquid (broth) mating**: Donor and recipient cells are mixed in a liquid culture (with or without shaking) and incubated together for a period (typically several hours) [38]. Because cells are free in solution, mating pairs form less frequently, so liquid matings usually give lower conjugation frequencies than solid matings. However, liquid mating can be used (sometimes with low agitation) for certain strains or plasmids.

In all cases, after mating, the cultures are diluted and plated on selective media that allows only transconjugants (and, if needed, the donor or recipient) to grow.

Pioneering studies in the 1980s showed that LAB—previously considered “transformation-resistant”—can receive broad-host-range plasmids via conjugative transfer. Vescovo’s group tested the transfer of the 17 MDa plasmid pAMβ1 (carrying erythromycin and lincomycin resistance) into *Lactobacillus* species [40]. Their objective was to circumvent the difficulty of transforming these *lactobacilli* by using conjugation. Using a filter-mating protocol, they mixed a donor (likely an Enterococcus carrying pAMβ1) with *L. acidophilus*, *L. reuteri,* and *L. salivarius* on solid medium. The methodology relied on selecting transconjugants on antibiotic plates to confirm plasmid acquisition. They observed that all three *Lactobacillus* species acquired pAMβ1, as evidenced by a newly detected plasmid band in transconjugants. This result confirmed successful conjugation and established that broad-host-range Gram-positive plasmids can infiltrate LAB by mating. Shrago extended this work by transferring pAMβ1 from *Streptococcus faecalis* into *Lactobacillus plantarum* [41]. They further showed that once *L. plantarum* harbored pAMβ1, it could serve as a donor to other *L. plantarum* strains, and *Streptococcus sanguis* could even be transformed with pAMβ1 extracted from *L. plantarum* and then conjugatively transfer it back to *L. plantarum*. The methodology involved sequential steps of inter-genus filter matings and plasmid isolations. The key finding was that pAMβ1 remained intact and transmissible across genera, demonstrating a truly broad host range. These early studies concluded that conjugation is a viable method to introduce genetic traits into LAB, overcoming natural transformation barriers.

Conjugative plasmids and ICEs in LAB encode conserved transfer functions, but their efficiency can vary by host. For instance, the enterococcal plasmid pIP501 (another broad-host-range replicon) and its relatives have been shown to transfer into *lactococci* and *lactobacilli* [42,43]. In *Lactobacillus sakei*, intergeneric transfer of pAMβ1 and pIP501 was achieved from donors like *E. faecalis*, although frequencies were not uniform across strains [44]. A standardized protocol for conjugation in dairy LAB (e.g., *lactococci*) was proposed in 2008 to assess antibiotic resistance transfer, emphasizing consistent donor–recipient ratios and mating times. Generally, filter mating on a nutritious agar surface is the common protocol: donor and recipient cultures are mixed (often at a 1:1 ratio), concentrated on a membrane or plate, and incubated to allow mating-pair formation. After several hours (or overnight), cells are plated on selective media that only allow growth of transconjugants (typically selecting for an antibiotic resistance carried by the plasmid and against the donor by exploiting host-specific antibiotic sensitivities). For example, a recent study by Samperio and colleagues adapted a Gram-negative mating protocol to transfer plasmids from *E. coli* into *Lacticaseibacillus casei*. They mixed overnight donor and recipient cells in equal volumes on a BHI agar (chosen because it supports both *E. coli* and *L. casei* growth, unlike LB or MRS which favor one or the other) [45]. After 24 h at 37 °C, cells were plated on selective media to recover *L. casei* transconjugants while counter-selecting *E. coli*. This protocol yielded measurable conjugation frequencies.

Conjugation frequencies in LAB can range from very low (~10^−6^ per donor) to relatively high (~10^−4^ or higher) depending on the system. Samperio’s work (2021) reported that using *E. coli* donor strain S17-1 (which carries the RP4 transfer machinery) to mobilize a shuttle vector into *L. casei* gave ~1.8 × 10^−3^ transconjugants per donor [45]. This was only about one order of magnitude lower than conjugation into *E. coli* itself, highlighting the effectiveness of the RP4 system in bridging Gram-negative to Gram-positive transfer. They also tested an R388-derived system and observed ~2.1 × 10^−4^ frequency in *L. casei*, versus ~2.5 × 10^−3^ in *E. coli*. Thus, with proper machinery (oriT on the plasmid and a compatible transfer apparatus in the donor), *E. coli* can serve as a DNA conduit into LAB with only modest drop in efficiency. However, success can vary greatly among LAB strains. The same study evaluated multiple wild *L. casei* isolates as recipients: all strains yielded transconjugants with the RP4-based donor, whereas the R388 system worked in only a subset of strains. Conjugation frequency also varied significantly among strains and was consistently lower in wild isolates compared to the laboratory strain used for optimization. This implies that factors like restriction enzymes, cell envelope differences, or recipient plasmid compatibility can impact conjugation. In practice, one often must optimize conditions (mating time, growth phase, donor–recipient ratio, and media) for each new donor–recipient pair. For example, increasing mating duration to 24 h and using a rich, non-selective mating medium improved yields in the *L. casei* protocol [45].

Beyond plasmids, conjugation has been harnessed to introduce transposons into LAB for random mutagenesis. Several research groups developed strategies of inter- and intraspecies deliveries via conjugation to LAB [45,46,47]. In such protocols, the donor harbors a suicide plasmid (unable to replicate in the LAB) with a transposon and the necessary conjugation functions. Mating results in one-way transfer of the transposon into the LAB chromosome (selected by antibiotic resistance on the transposon). The conclusion from those studies is that conjugation can bypass the need for competent cells or protoplasting, enabling genetic tools (plasmids, transposons, integrative vectors) to be delivered even into recalcitrant LAB strains.

Recent research has significantly expanded our understanding of conjugation in LAB, both in terms of natural plasmid transfer and engineered systems. In 2024, a detailed conjugation protocol was established for *Lactobacillus delbrueckii* subsp. *bulgaricus* using the pGMβ1 shuttle vector mobilized from *E. coli* [48]. This method achieves transfer frequencies of ~10^−3^ transconjugants per recipient without requiring electroporation, relying instead on filter mating at 30 °C for 6 h in GM17 medium supplemented with Mg^2+^. Such streamlined approaches highlight that conjugation can be as efficient as electroporation, while avoiding the need for making cells electrocompetent. Furthermore, *Lactococcus lactis* dairy strains continue to be a focus of conjugative studies. A 2024 genomic survey of *L. lactis* revealed at least 83 distinct ICEs, many of which carry important traits such as lactose/raffinose metabolism or bacteriophage resistance, demonstrating that conjugation not only serves as a genetic tool but also shapes the functional diversity of industrial LAB populations [49].

Innovative conjugation-based tools have also emerged. In 2025, the “XPORT” platform was introduced—an inducible conjugative plasmid system designed for safe and controlled transfer into LAB [50]. The core design deletes an essential transfer gene (e.g., trbC or trbF, key components of the mating-pair formation/Type IV secretion system) and rescues it from an inducible promoter (P_BAD_; L-arabinose-inducible). In the OFF state (no arabinose), the plasmid cannot assemble a functional transfer apparatus and does not conjugate; upon induction, expression of the missing gene reconstitutes the transfer machinery, enabling oriT nicking/relaxase-guided transfer and a time-gated burst of conjugation. The authors showed this works with the complement supplied either in cis (on the same plasmid) or in trans (on a helper), with trans limiting onward spread because recipients lack the complement. Induction produced up to ~10^5^-fold increases in transfer efficiency relative to uninduced controls, demonstrating stringent control of the conjugation window [50]. It should be noted that the “XPORT” name was originally used for an ICEBs1-based, inducible donor strain that separates the Type IV secretion system from the DNA cargo and uses an auxotrophic handle for recovery [51]. The Jaafar and colleagues implemented a plasmid-borne variant of the same idea—inducible, biocontained conjugation—via essential-gene control. Together they outline a general blueprint for on-demand, safer horizontal transfer that can be adapted to LAB toolchains.

Complementary to these experimental advances, recent studies underscore the prevalence of conjugative plasmids in LAB genomes. A 2024 analysis of *L. lactis* SK11 identified two large (~40 kb) conjugative plasmids encoding not only lactose/raffinose metabolism but also anti-phage defense genes, suggesting that beneficial phenotypes in starter cultures may be linked to transferable genetic elements [52]. Similarly, a 2025 genomic study of *Enterococcus lactis*—a LAB species isolated from goat feces—discovered a mobilizable Inc18-family plasmid and over 30 conjugation-related genes, reinforcing the idea that conjugation remains an active force in LAB evolution and safety assessment [53]. Together, these findings highlight how both natural and engineered conjugation systems are now central to LAB biotechnology, providing reliable alternatives to classical transformation techniques and extending genetic accessibility to previously recalcitrant strains.

## 5. Transduction of LAB

Transduction is the process by which bacteriophages shuttle DNA from one bacterium to another. There are two types of transduction: generalized transduction, where a lytic phage accidentally packages random host DNA (plasmid or chromosomal fragments) into phage capsids, and specialized transduction, where a temperate phage erroneously excises with adjacent host genes. In LAB systems, both types have been documented, though examples are limited to specific phage–host pairs. This process is schematically shown in Figure 4.

In the late 20th century, evidence emerged that *Lactococcus lactis* could undergo phage-mediated gene transfer. Birkeland and Holo (1993) created a model system with lactococcal phage ΦLC3 to deliberately transduce a plasmid [54]. They cloned the phage’s cohesive end site (cos; cohesive end sequences used by λ-like phages for genome packaging) into a plasmid, allowing phage packaging of the plasmid DNA during infection. Using *L. lactis subsp. cremoris* as both donor and recipient, they induced the temperate phage ΦLC3 in a donor harboring the cos-containing plasmid. The phage particles that emerged often encapsulated the plasmid (instead of phage genome) and could infect a new *L. lactis* host. Transductants were obtained at frequencies of ~10^−4^–10^−3^ per phage particle [54], depending on plasmid size (smaller inserts yielded higher packaging frequency). The key conclusion was that the presence of a suitable cos sequence enabled efficient plasmid transduction, and that packaging capacity was the limiting factor (larger DNA inserts lowered the titer of transducing particles (transducing units per PFU)). This work highlighted that phage DNA packaging signals can be exploited to ferry non-phage DNA into LAB. However, the system was limited to strains permissive to that phage and required the plasmid to carry specific phage sequences, making it a specialized laboratory tool rather than a general method.

Around the same time, researchers observed rare generalized transduction events in *lactococci* using lytic dairy phages. For instance, Schouler reported phage bIL67 could transduce lactose-fermenting ability between *L. lactis* strains (via transfer of a lactose plasmid) [55]. These frequencies were very low, often detectable only by sensitive selection (on the order of 10^−7^–10^−8^ per PFU, several orders of magnitude below conjugation or electroporation). Still, the proof-of-concept was important: it showed that lytic phages infecting LAB sometimes package host DNA by mistake. Factors like phage DNA packaging mechanisms influence this: phages with pac sites (packaging initiation sequences; these phages initiate packaging at a specific sequence and then cut DNA into headful lengths) versus cos sites (which package only genomes with a particular cohesive end) differ in how likely they are to grab host DNA. In *Lactococcus*, cos-type phages (e.g., λ-like phages) can be harnessed by adding cos sites to vectors (as Birkeland did), whereas pac-type phages might pick up any DNA if it is concatemerized (which indicates head-to-tail multimers that can be captured by pac-type systems) and a pac sequence is recognized [56,57].

A breakthrough came from studies of *Lactobacillus delbrueckii* phages. Ravin in 2006 discovered that the virulent phage LL-H, which uses a pac-type packaging mechanism, could transfer a particular plasmid pX3 at astonishingly high frequency [56]. Their experimental aim was to evaluate phage LL-H’s propensity to carry host plasmid DNA. They infected *L. delbrueckii* strains containing plasmid pX3 and observed that after phage replication, most progeny phage particles packaged plasmid DNA concatemers instead of phage DNA. The transduction frequency of pX3 reached 0.1 to ~1 transducing particle per PFU (plaque-forming units), meaning up to 100% of the phage population carried the plasmid [56]. By contrast, a control plasmid (pJK650) lacking certain sequences was transduced at only ~2 × 10^−2^ frequency. The methodology involved treating the donor culture with a DNA-damaging agent to induce phage production, then filtering the lysate and infecting recipient *L. delbrueckii* cells. Transductants were selected by the antibiotic resistance encoded on pX3. Molecular analysis revealed that pX3 had a fortuitous “pac-like” site ~1 kb from one end, which likely tricked the phage packaging machinery. Moreover, pX3 presence in the cell seemed to interfere with phage genome replication, so the phage was forced to package plasmid multimers. The outcome was a high-frequency, nearly pure population of transducing particles. This was a remarkable result: it demonstrated a natural, highly efficient transduction system in LAB. The authors concluded that phage LL-H + pX3 could serve as a powerful DNA delivery tool for *L. delbrueckii*. Practically, this means one could use phage LL-H to move any gene of interest cloned into the pX3 vector between *Lactobacillus* strains with unprecedented efficiency. No analogously efficient phage transduction system exists in *E. coli* (where P1 transduction frequencies are typically 10^−5^–10^−6^ at best), highlighting how unique this phage-plasmid interaction is.

Another notable example is the temperate phage Φadh of *Lactobacillus gasseri*. Damelin’s work leveraged Φadh to deliver an engineered plasmid into *L. gasseri* for protein expression [58]. Their objective was to create a *Lactobacillus* that produces human chemokine proteins without using traditional transformation. They induced an *L. gasseri* ADH donor strain carrying a shuttle plasmid (with the chemokine gene and an antibiotic marker) to trigger Φadh’s lytic cycle (using mitomycin C). The released phage particles, some of which encapsidated the shuttle plasmid, were used to infect a plasmid-free *L. gasseri* recipient. Transductants were obtained at high frequency, implying efficiencies on par with or better than electroporation for that strain [58]. Importantly, this method does not require the target strain to be competent or electrocompetent—the phage naturally injects the DNA. Damelin and colleagues confirmed that transduced *L. gasseri* expressed the CC chemokine from the plasmid, illustrating phage-mediated gene delivery for functional protein production. This system was heralded as the first high-efficiency transduction toolkit for *Lactobacillus*, and it showcased a protocol: induce a lysogenic phage in the donor (with or without UV/chemical induction), filter sterilize the lysate, then adsorb it to the recipient culture and plate on selective media after an incubation for infection. The study’s conclusion was that temperate LAB phages can be “hijacked” for genetic engineering, providing an alternative when standard transformation fails. Indeed, spontaneously induced Φadh-like transducing particles were later shown to contribute to horizontal gene transfer in *L. gasseri* even without deliberate induction, as *L. gasseri* cultures naturally release phage at ~10^7^ PFU/mL in the stationary phase [59].

An intriguing finding by Ammann was that phages of *Streptococcus thermophilus* (a yogurt bacterium) could transduce DNA into *Lactococcus lactis*, crossing genus boundaries [57]. They demonstrated plasmid transfer from *S. thermophilus* to *L. lactis* via phage, albeit at low frequencies. Specifically, they used virulent *S. thermophilus* phages on a donor carrying a lactose-fermentation plasmid, and recovered *L. lactis* recipients that acquired the plasmid trait. The transduction frequency was about 4 orders of magnitude lower than typical intra-species transduction, but it occurred. This result is significant as it suggests phages might mediate gene flow between different LAB species in dairy environments (where both *S. thermophilus* and *L. lactis* might be present, such as in mixed starter cultures). It also implies that the host range of some phages can be broad enough to infect related LAB genera, opening possibilities for phage-mediated DNA delivery beyond a single species. Ammann concluded that while cross-species phage transduction is rare, it sets a precedent for using phages to mobilize desirable plasmids across LAB species. For genetic engineering, this hints that a plasmid could be “moved” from one LAB strain to another by using a phage that infects both, without direct human manipulation of the DNA.

Despite these successes, phage transduction is not routine for most LAB. One limitation is host range—phages tend to have a narrow range of hosts. A phage that works for *L. lactis* MG1363 will not infect a *Lactobacillus* strain and may even fail on a different *L. lactis* strain. Bron and colleagues note that conjugation and transduction are constrained by host compatibility and DNA size packaging limits [36]. Another issue is the need to avoid killing the recipient: truly lytic phages destroy the host cell after injecting DNA, so any transduced DNA must either establish before the cell lyses or come from defective phage particles. Approaches like using defective transducing particles (as in the ΦLC3 cos-system or the pac-heavy LL-H system) mitigate this, since those particles inject DNA without a full phage genome to continue infection. Temperate phages can also be used under conditions where they infect and establish lysogeny carrying new genes, but temperate phages often repress or integrate rather than maintain plasmids. In practice, transduction protocols might require tedious screening to find transductants among many phage-infected (and possibly killed) cells.

In *E. coli*, generalized transduction (using phage P1) and specialized transduction (using λ) are classical tools—P1 can move virtually any chromosomal gene between *E. coli* strains at ~10^−5^ frequency, and λ-derived vectors can integrate/excise specific loci. LAB lack a universal transducing phage like P1. The phages are more diverse and often less amenable to broad use. *B. subtilis* has phage SPP1 or PBS1 for generalized transduction, but even these are less frequently used than natural transformation in that species. LAB, on the other hand, often cannot be naturally transformed (with notable exceptions like *Streptococcus thermophilus* under special conditions) and may be poor candidates for electroporation (especially wild isolates). For those strains, phage transduction becomes a compelling option. As Marcelli put it, they sought to provide a proof of concept that bacteriophage transduction may be used in an industrial setup to mobilize plasmid and chromosomal DNA among *L. lactis* strains as an alternative genetic improvement strategy [60]. In their work, they screened strictly lytic lactococcal phages for ability to transduce markers. They succeeded in transducing two different plasmids between *L. lactis* strains using three unrelated phages. The frequencies were modest but detectable, and importantly, no antibiotic resistance or foreign DNA was introduced beyond what a phage naturally moved. This aligns with a regulatory perspective: modifications achieved by phages might be viewed as “natural” since they mimic HGT events that could occur in fermenters spontaneously. In essence, conjugation and transduction represent a renaissance of traditional DNA transfer strategies in LAB biotechnology, enabling researchers to genetically modify food-grade strains without resorting to transgenic methods that involve antibiotic markers or exogenous DNA assembly.

Phage transduction in LAB, while not as universal as in model bacteria, has proven feasible in several contexts. It has been used to transfer plasmids encoding metabolic traits, bacteriocin production, and even antibiotic resistance genes between LAB. Successful transduction has been reported in *Lactococcus lactis*, *Lactobacillus delbrueckii*, *Lacticaseibacillus casei* (via induced prophage), and others, including *Streptococcus thermophilus*. Each system requires identifying or engineering a phage that can package the DNA of interest and infect the target strain. The methodologies typically involve inducing a donor bacterium’s phage and using its lysate to infect the recipient under conditions where only transductants will survive (e.g., selective media). When applicable, these methods complement conjugation and electroporation. For LAB strains recalcitrant to other transformation techniques, conjugation and transduction offer vital alternatives. They also deepen our understanding of horizontal gene transfer in LAB populations. Going forward, improved insight into the molecular mechanisms (e.g., the DNA packaging specificity, receptor range, and restriction evasion) should further increase the reliability of LAB conjugation and transduction. Such advances will bring LAB engineering closer to the ease of *E. coli*, while maintaining the “food-grade” and natural status that is often required in probiotic and food industry applications.

## 6. Biolistic Particle Delivery

Biolistic transformation (microprojectile bombardment) is a physical DNA delivery method in which DNA-coated particles (often gold or tungsten microbeads) are accelerated into target cells. This technique was first developed in the late 1980s for plant genetic engineering—Klein and colleagues invented the “gene gun” to overcome host-range limits of *Agrobacterium* vectors [61]. The approach was soon adapted to other kingdoms; for example, the first bacterial biolistic transformation was demonstrated in *Bacillus megaterium* (a Gram-positive bacterium) by shooting plasmid-coated microprojectiles into cells [62]. Because it bypasses biological uptake mechanisms, biolistics can in principle deliver DNA to virtually any cell type given appropriate projectile size and velocity. Historically, however, it remained a niche method due to technical complexity and cell damage issues.

In practice, biolistic transformation is seldom used in bacteriology, but it has seen specific application with Gram-positive microbes—especially when conventional methods fail. Most lactic acid bacteria (LAB) are more routinely transformed by electroporation or conjugation, so reports of gene gun use in LAB are rare [63]. Notably, biolistics has proven valuable for certain recalcitrant LAB strains. A recent breakthrough demonstrated stable plasmid transformation of *Oenococcus oeni* (a malolactic fermentation LAB) using a gene gun: Chen and Chen achieved reproducible introduction of a shuttle vector into *O. oeni* by bombarding cells with detonation nanodiamond particles coated in DNA [64]. This result is significant because *O. oeni* had long resisted transformation by standard means—even a plasmid vector built with *O. oeni*’s own replicon failed to yield transformants via electroporation [65]. The success of the gene gun in this case highlights how physical bombardment can overcome extreme permeability barriers or restriction defenses in Gram-positive cells. In fact, the challenge of penetrating thick Gram-positive cell walls was recognized early on: using finer (sub-micron) gold particles in a particle-inflow gun, Elliott in their work managed to transform *Bacillus subtilis* via biolistics [66]. Today’s use of nanoscale carriers—e.g., ~5 nm diamond particles in *O. oeni*—follows the same principle of miniaturizing projectiles to improve DNA delivery into small or robust cells. Overall, while not common, gene gun methods are being revisited in the genomics era as a tool of last resort for genetically intractable Gram-positive bacteria.

Compared to standard LAB transformation techniques, biolistic delivery has distinct pros and cons. It is broadly applicable—not requiring specific host receptors, mating-pair formation, or phage infection—so in theory one gene gun can service many species. This contrasts with conjugation and transduction, which depend on biological compatibility (presence of a transferable plasmid or a susceptible phage) and are often limited to certain hosts. However, biolistic transformation’s practicality is limited by its specialized equipment (helium-driven gun or similar devices) and typically lower efficiency. Studies note that electroporation usually yields far higher colony counts than particle bombardment under comparable conditions [67]. Moreover, blasting cells with microprojectiles can cause high mortality in the target population, whereas conjugation and transduction, when available, tend to be gentler on cells and can achieve high transfer frequencies in their compatible hosts. In LAB research, optimized electroporation remains the primary method for most strains due to its simplicity and relatively high success rate, while conjugative transfer and phage-mediated transduction are useful alternatives for certain cases [63]. Biolistics, by comparison, is reserved for special scenarios—it can circumvent otherwise intractable barriers where other methods falter, but it is not the first-line choice for routine genetic manipulation of LAB. When deployed strategically (e.g., to bypass an unbreachable cell envelope or to avoid host-range restrictions), the gene gun method provides a valuable complementary approach alongside classical transformation, conjugation, and transduction in the molecular toolkit for LAB. A comparison for all discussed methods of transformation of LAB is presented in Table 2.

## 7. Native and Engineered Plasmids of LAB (Shuttle Vectors)

LAB, including genera like *Lactococcus* and *Lactobacillus*, naturally carry numerous plasmids. For example, *Lactococcus lactis* strains often harbor 4–7 plasmids ranging from ~0.9 kb up to >80 kb in size [68,69,70,71,72]. These native plasmids encode a variety of functions beneficial for dairy and probiotic applications, such as lactose and citrate fermentation, bacteriocin production, heavy metal resistance, stress tolerance, and phage resistance [72,73,74,75,76,77,78,79,80,81,82,83,84,85,86,87,88,89,90,91,92,93]. Early studies revealed that LAB plasmids utilize two main replication modes—rolling-circle (RCR) and theta—which influence their properties. RCR plasmids (common in Gram-positives) tend to have broad host range and high copy number, since they rely on a simple Rep protein and minimal host factors. However, the single-stranded DNA intermediate in RCR replication makes them structurally and segregationally unstable, especially for large inserts [94]. In contrast, theta-replicating plasmids (which replicate via double-stranded intermediates) usually have lower copy numbers but high stability and can accommodate larger DNA fragments [95]. This inherent difference means that many native LAB plasmids under ~10 kb use RCR, whereas larger plasmids use theta mechanisms, often including partition systems for faithful segregation.

Researchers have leveraged these native plasmids to develop a wide array of cloning and expression vectors for LAB over the past few decades [68,96,97,98,99,100,101,102,103,104,105,106,107,108]. Engineered LAB plasmids typically combine a LAB-derived replicon with features familiar from *E. coli* vectors (multi-cloning sites, selection markers), sometimes creating shuttle vectors that replicate in both LAB and *E. coli*. Because typical *E. coli* origins (e.g., ColE1) do not function in Gram-positive hosts, shuttle plasmids carry dual origins. For instance, the shuttle vector pLES003 was constructed with a replication origin from a *Lactobacillus brevis* plasmid (pLB925A03) plus a ColE1 origin and pUC19-derived cloning region [104]. This design allows replication in *E. coli* for easy DNA manipulation, as well as in multiple LAB species (pLES003 is stable in *L. brevis*, *L. plantarum*, *L. helveticus*, *Enterococcus hirae*, etc.) [104]. Another example is pGYC4α, derived from an *L. sakei* RCR plasmid, which was shown to express a *Bacillus* α-amylase gene not only in *E. coli* but also in *L. lactis* and even *Leuconostoc* species [105,106]. These successes highlight that LAB replicons can be paired with Gram-negative origins to create versatile shuttles for cloning and protein expression across domains.

Early LAB vectors often utilized small, high-copy RCR replicons (for convenience and yield), but those proved less suitable for large or industrial constructs due to instability. Studies found that RCR-based vectors could only stably carry relatively small inserts (often <5–10 kb)—larger DNA payloads led to deletions or plasmid loss [94]. This is because multimer formation and random segregation of single-stranded intermediates make high-copy plasmids prone to recombination and curing in the absence of selection [94,95]. To address this limitation, researchers shifted toward theta-replicating plasmids for vector development [96]. Theta-type replicons, often derived from medium-sized cryptic plasmids, confer lower copy number and greater stability. For example, Panya and colleagues isolated two theta plasmids (~2.9 kb and 13.9 kb) from *Lactobacillus casei* and used their replicons to build a new shuttle vector series that was stably maintained even across different LAB species [107]. More recently, a large plasmid pMC11 (~30 kb) from *L. casei* was found to carry two distinct theta replicons [108]. Engineering each replicon into a shuttle vector yielded pEL5.7 and pEL5.6, which, in turn, were used to construct expression plasmids pELX1 and pELX2. These plasmids successfully expressed GFP in multiple *Lactobacillus* strains, demonstrating strong inter-species functionality. Such robust theta-based vectors are valuable for LAB genetic engineering, as they combine adequate capacity with reliable maintenance.

Another critical aspect of plasmid engineering is the selection marker. Native LAB plasmids often carry genes for antibiotic resistance (e.g., erythromycin, chloramphenicol resistance) or metabolic functions that can be exploited as selectable traits. Accordingly, many LAB cloning vectors have traditionally used antibiotic resistance for selection [96]. For instance, the popular nisin-inducible vectors (NICE system) use chloramphenicol or erythromycin resistance genes for maintenance. However, in food and probiotic applications, the use of antibiotic markers is undesirable or even prohibited. This motivated the development of food-grade selection systems that avoid antibiotic genes. One strategy uses complementation of auxotrophic mutants: for example, an alanine racemase gene (*alr*) can be the selection marker on a plasmid, allowing only *Lactobacillus plantarum* cells with an *alr* deletion to grow when alanine is absent [101]. Another approach uses bacteriocins and immunity genes as selectable traits. In this strategy, the plasmid encodes an immunity protein that protects the host from a bacteriocin that is added or produced; cells that lose the plasmid are killed by the bacteriocin. This has been demonstrated with nisin, lacticin 481, and lacticin 3147 systems in *L. lactis* [109,110,111]. Recently, the lactococcin 972 (Lcn972) bacteriocin operon was adapted as a novel post-segregational killing system: when the Lcn972 immunity and production genes are carried on a plasmid, cured cells are selectively eliminated, allowing stable plasmid maintenance without antibiotics [76]. Researchers reported that an *L. lactis* strain harboring an Lcn972-based vector retained the plasmid without antibiotic pressure and without any noticeable fitness cost, a promising development for safe industrial use.

In summary, LAB plasmid engineering has yielded a toolkit of vectors derived from native plasmids, balancing factors like copy number, host range, and selection strategy. Modern LAB shuttle vectors incorporate small Gram+ replicons (often theta-type for stability) combined with an *E. coli* origin for easy handling. These plasmids have enabled cloning and high-level expression of enzymes and antigens in LAB, analogous to *E. coli*’s plasmid systems but tailored to LAB biology. A large number of examples exist—from the pSIP series (based on the *L. sakei* sakacin pheromone plasmid) used for high-yield protein production in *Lactobacillus*, to the pNZ series in *L. lactis* derived from cryptic dairy plasmids. Notably, pSIP demonstrates how a native quorum-sensing mechanism (induction by a secreted peptide) was harnessed for a regulatable expression vector [97]. The only drawback was its erythromycin resistance marker, which spurred efforts to swap in food-grade markers. This exemplifies a broader trend: LAB plasmid systems increasingly use endogenous LAB genetic elements (origins, immunity genes, metabolic markers) to remain “food-grade”. Comparisons with model bacteria highlight these adaptations: unlike *E. coli* plasmids, LAB vectors must function in low-GC, restriction-proficient hosts and often in rich fermentation media. By leveraging native plasmid biology, scientists have created LAB plasmid vectors that are effective tools for both laboratory research and industrial biotechnology, bridging the gap between Gram-positive and Gram-negative molecular genetics. Table 3 presents a quick overview of the vectors and vector families that are discussed below in more detail.

## 8. Suicide Plasmids of LAB: Integration Vectors and Gene Knockout Systems

Genetic engineering of LAB (to knock out genes or insert new DNA) has historically relied on “suicide” plasmids, which are vectors that cannot replicate in the target LAB host. When introduced into a host where they cannot self-replicate, such plasmids can only be maintained by integrating into the chromosome via homologous recombination. This strategy ensures that antibiotic selection for the plasmid leads to stable chromosomal insertion of the plasmid’s cargo (often disrupting a target gene or introducing a new gene into the genome). Several clever systems were developed in the 1990–2000s to accomplish this in LAB, a time when tools like CRISPR or Lambda-Red (common in *E. coli*) were not available.

A pioneering example is the *Lactococcus lactis* pORI/pVE6007 integration system developed by Law and colleagues [116]. In this two-plasmid system, a vector called pORI19 provides an *L. lactis* origin of replication (ori) but lacks the repA gene required for replication initiation. On its own, pORI19 (Ori+ RepA–) behaves as a suicide plasmid in *L. lactis*—it cannot propagate there because RepA (the replication protein) is missing. However, pORI19 can be propagated in a specialized *E. coli* strain (EC101) that supplies the *Lactococcus* RepA in trans, allowing cloning of DNA fragments into pORI19 in *E. coli*. The role of the cloned DNA is to provide homology to a target locus in *L. lactis*. To perform the integration, *L. lactis* is first cured of any replicating RepA source (ensuring that pORI19 cannot autonomously replicate when introduced). The recipient *L. lactis* is instead given a helper plasmid, pVE6007, which is a RepA+ vector carrying a temperature-sensitive (Ts) replication origin [116]. At a permissive temperature (around 28–30 °C), pVE6007 produces RepA, allowing a pORI19 recombinant plasmid to replicate transiently after transformation. The presence of RepA and low temperature thus makes *L. lactis* maintain pORI19, giving time for recombination to occur. Subsequent shifting of the culture to the non-permissive temperature (37 °C) in the absence of RepA selection causes pVE6007 to be lost (its Ts replicon cannot function at 37 °C) and simultaneously forces pORI19 to either integrate into the chromosome or be lost. Recombination between the cloned fragment on pORI19 and the identical chromosomal sequence results in single-crossover (Campbell-type) integration, inserting the entire plasmid into the genome. Using this method, Law in their work constructed a library of chromosomal insertion mutants in *L. lactis* and could readily screen for phenotypes [116]. They successfully disrupted genes such as *acmA* (a lactococcal peptidoglycan hydrolase) and a gene in the maltose metabolism pathway, confirming integrations by PCR and phenotype. Moreover, because the integration was reversible (a second crossover can excise the plasmid), the system allowed recovery of mutations or allele replacements by reintroducing pVE6007 at permissive temperature to “rescue” the plasmid from the chromosome. This pORI19/pVE6007 toolkit was a breakthrough for *L. lactis*, enabling stable gene knockouts without permanent vector replication in the final strain.

The same RepA-based suicide strategy was later adapted to other LAB. Russell and Klaenhammer (2001) developed an efficient integration method for *Lactobacillus acidophilus* and *L. gasseri* by modifying the lactococcal pORI system [120]. Their approach similarly used a replication-deficient vector (carrying *Lactobacillus* homology) and a helper plasmid for conditional replication. They demonstrated site-specific insertional inactivation of the *L. acidophilus lacL* gene (encoding β-galactosidase) and the *L. gasseri gusA* gene (a β-glucuronidase) by selecting for mutants that had integrated the suicide vector into these loci [120]. The success in *Lactobacillus* showed the broad applicability of conditional replication systems: by supplying or withholding a key replication protein, one can toggle a plasmid between a replicative state and a suicide (integrative) state. Notably, Leenhouts also described variants of pORI vectors for *Lactococcus* that allow unmarked gene replacements, where a second recombination event excises the plasmid leaving behind only a designed mutation (with no antibiotic resistance genes) [117]. Such two-step integration–excision processes became a standard for generating GRAS (generally recognized as safe) modified LAB strains with no residual foreign DNA.

An alternative class of LAB suicide vectors exploits temperature-sensitive replication in a single-plasmid approach. Rather than requiring a separate helper, these plasmids carry a mutant replicon that is functional at a lower temperature (typically 30 °C) but nonfunctional at a higher temperature (37–40 °C). One example is the pG + host9 vector developed by Maguin and colleagues [118]. This plasmid is based on the *L. lactis* pWV01 replicon, with mutations that render RepA inactive at high temperature. At 30 °C, pG + host9 replicates in Gram-positive hosts like *Lactococcus* or *Lactobacillus*; when shifted to 37–39 °C, the plasmid can no longer replicate and is rapidly lost from dividing cells unless it integrates into the chromosome. Serror in 2003 demonstrated this behavior: in *Lactobacillus delbrueckii*, after 24 generations at 44 °C (non-permissive), only ~0.1% of cells retained the pG + host9 plasmid, whereas at 37 °C (near the permissive threshold) ~100% still retained it [119]. This shows that pG + host9’s replication is tightly temperature-regulated, making it essentially a suicide vector at elevated temperatures. By cloning either homologous chromosomal sequences or even transposons into such Ts plasmids, researchers achieved random or targeted insertional mutagenesis in LAB. For instance, pG + host9 carrying the ISS1 insertion sequence was used to generate random mutations in *L. lactis* through a “pop-in” transposition mechanism: the plasmid would transpose ISS1 into the genome and then be cured at high temperature, leaving behind a single copy of ISS1, disrupting a gene [119,121]. Thousands of such mutants were isolated to study gene functions (this was an early form of transposon mutagenesis in LAB).

Overall, suicide plasmids have been indispensable for LAB genetics, enabling chromosomal gene modifications prior to the advent of more modern genome-editing tools. The key design principle is to include a LAB-targeted DNA segment (for homologous recombination or transposition) on a vector that cannot independently propagate in that LAB. *E. coli* analogs of this strategy exist (e.g., suicide plasmids for Gram-negatives that replicate only in specific hosts or require pir+ strains), but LAB-specific challenges (like difficulty of natural transformation and strong restriction–modification systems) made the development of such tools a significant accomplishment. LAB suicide vectors like pORI19 and pG + host9 are often used in tandem with selection–counterselection schemes (e.g., using a sacB gene for sucrose sensitivity) to generate clean knockouts. When comparing to *Bacillus subtilis*, we see a similar use of Ts plasmids (e.g., pE194) for integration, underscoring that conditional replication is a common solution in Gram-positive bacteria for allele replacement. The success of these systems also highlights LAB’s difference from *E. coli*: because most LAB plasmids are not conjugative or high-frequency mobile elements, researchers had to engineer replicons and clever host backgrounds to achieve the necessary recombination events. Today, with these foundational tools, one can integrate new metabolic pathways into *L. lactis* or delete virulence factors from *Lactobacillus* probiotics by simply designing the appropriate suicide plasmid and leveraging the temperature or helper-plasmid controls to insert or remove DNA at will.

## 9. Plasmids with Regulated Replication in LAB. Temperature-Sensitive & Inducible Origins

Controlling a plasmid’s replication—turning it “on” or “off”—is a powerful approach in bacterial genetics, allowing temporal regulation of gene dosage or facilitating plasmid curing. In LAB, the most widely used regulated replication systems are those with temperature-sensitive (Ts) origins. A Ts plasmid replicates only under permissive temperature conditions and is unable to initiate replication at higher (non-permissive) temperatures. We described this concept with the pVE6007 helper and the pG + host9 vector in the previous section. Ts plasmids can be considered a subset of suicide plasmids, but they are notable for providing a reversible, conditional control: one can maintain the plasmid at 30 °C and then inactivate it by shifting to 37–40 °C. This property is extremely useful not only for chromosomal integration (forcing a single crossover as described) but also for plasmid retention studies and plasmid curing. For instance, to remove an antibiotic-resistance-bearing plasmid from a LAB strain after genetic manipulation, a Ts replicon can be used so that simply growing the strain at elevated temperature—without antibiotics—will selectively lose the plasmid in nearly all cells [119]. Because LAB are often grown at 30 °C for laboratory purposes and 37 °C for animal host studies, Ts plasmids offer a convenient toggle.

The creation of Ts LAB plasmids typically involves point mutations in the replication protein or origin. In the case of pWV01 (a common lactococcal replicon), a mutant RepA was isolated that was functional at ~28 °C but inactive at ≥37 °C. This mutant RepA(Ts) was used in vectors like pVE6007 and later pG + host9 [116]. Similarly, researchers have applied Ts mutations to broad-host-range plasmids from other Gram-positives. For example, the enterococcal theta plasmid pAMβ1 has a replication machinery that, when altered, can show temperature dependence (some class D theta replicons are known to have temperature-labile variants) [116,118]. One comprehensive study in *Lactobacillus delbrueckii* compared the stability of a pG + host RCR plasmid vs. a theta plasmid pIP501 at different temperatures: both were found to be temperature-sensitive in that host, disappearing at high temperature without selection [119]. This indicates that Ts replication can sometimes be host-dependent; a plasmid might be stably maintained in one LAB species but not in another at the same temperature, due to differences in host replication factors or chaperones. Thus, Ts vectors often need to be empirically tested in the target LAB. Nonetheless, they remain the go-to method for regulated replication in LAB. By adjusting growth temperature, researchers can modulate plasmid copy number to some extent (permissive temperature often yields higher copy, semi-permissive yields lower copy), or eliminate the plasmid entirely, which is useful for in vivo applications where the plasmid should not persist.

Beyond temperature, are there inducible replication origins in LAB? Compared to Ts systems, direct chemical control of plasmid replication is less common in LAB. In *E. coli*, one can find plasmids with regulated copy number (for example, the RK2 plasmid’s copy control system or engineered origins that respond to an inducer by antisense RNA regulation). In LAB, no widely adopted single-plasmid system exists where adding a small molecule (like IPTG or an antibiotic) initiates or halts replication. However, the two-component pORI system discussed earlier can be viewed as an “inducible origin” system in a broader sense—the presence of RepA (supplied by a helper plasmid) induces replication, and removal of RepA stops replication. One could imagine placing the repA gene under an inducible promoter on a separate replicon: in that scenario, the addition of an inducer (say nisin or IPTG to express RepA) would allow a RepA– plasmid to start replicating. While this specific configuration has not been reported as a standard tool, it is conceptually similar to how pORI/pVE6007 works (with temperature as the trigger for RepA availability). Thus, LAB researchers have achieved conditional replication mainly by controlling the presence of an essential replication factor or by leveraging temperature-labile DNA polymerase interactions.

It is worth noting that all plasmids have some form of replication regulation intrinsically. Many LAB plasmids, especially theta-type, encode copy-number control elements such as iteron sequences or antisense RNAs that ensure the plasmid does not over-replicate. For instance, the lactococcal plasmid pCI305 (theta type) and the broad-host plasmid pIP501 both have small antisense RNAs (RNAII/RNAIII systems) that titrate the replication initiator and keep copy number low and stable [114,122]. While these mechanisms are not externally induced by the researcher, they represent regulated replication from a biological perspective—the plasmid regulates itself in response to the host cell’s physiology. One practical upshot is that theta plasmids in LAB are usually maintained at 1–5 copies per cell (low-copy), which, as studies have shown, tends to improve the stability of protein expression and reduce metabolic burden. In fact, a 2020 study comparing expression from multi-copy vs. low-copy vectors in *Lactococcus* found that a single-copy integrative vector could outperform a high-copy plasmid for steady protein production [123]. This is analogous to findings in *E. coli* metabolic engineering: sometimes less is more, as a very high plasmid copy can stress the cell and lead to mutations or plasmid loss. Therefore, LAB engineers often choose a replicon appropriate for the task (e.g., a low-copy theta replicon for a stable long-term expression strain, versus a high-copy RCR replicon if trying to boost yields transiently).

Another consideration for using multiple plasmids in one LAB cell is plasmid incompatibility. If two plasmids share the same replication machinery (e.g., both rely on the pAMβ1 replicon or both on pWV01), they cannot be stably maintained together—one will outcompete or displace the other [124,125]. Thus, to co-express multiple genes, one must use plasmids from different incompatibility groups (different replicon families). This has led to the development of sets of vectors for LAB that have distinct replicons (for instance, pairing a pWV01-derived vector with an pCI305-derived vector in the same cell). Some researchers have managed modular multi-plasmid systems in *Lactococcus* to express up to three different plasmids simultaneously by careful replicon choice and selection, though this is nontrivial due to the limited number of fully orthogonal replicons. By contrast, *B. subtilis* usually integrates additional genes into its chromosome rather than maintain many plasmids, and *E. coli* can use plasmids of different incompatibility groups (ColE1 vs. p15A vs. F, etc.) with relative ease. LAB systems are catching up in this regard: the classification of LAB plasmid replicons into different classes (A–F) [124,126] helps identify which plasmids might coexist. For example, a class A theta replicon (like pCI305 family) can be combined with a class C RCR replicon (like pMV158 family) in one strain with minimal interference.

In summary, temperature-sensitive plasmids stand out as the main tool for regulated plasmid replication in LAB. They have been instrumental for conditional mutagenesis and curing, functioning as a sort of “plasmid switch” controlled by heat. While true chemically inducible origins are not part of the standard LAB toolkit, flexibility in plasmid copy number and maintenance can be achieved by other means: using different replicon types, employing helper plasmids for trans-replication, or simply by exploiting the host’s growth conditions (temperature, nutrients) that affect plasmid stability. The lessons from *E. coli* and *Bacillus* reinforce these strategies. *Bacillus* researchers routinely use Ts plasmids (e.g., Ts derivatives of pE194 or pTA1060) to integrate DNA or cure plasmids—a parallel to LAB. *E. coli*, being mesophilic, does not use Ts replication as often, but instead may use inducible copy number plasmids (like adding an inducer to increase pBR322 copy via a mutant Rop protein, etc.). LAB have not needed such fine chemical control yet; the combination of strong inducible promoters on a low-copy plasmid often suffices for expression tuning. Should a need arise for truly switchable plasmid maintenance in LAB (for example, a probiotic that carries a plasmid-encoded function which should activate only under certain gut conditions and then be lost), future designs might incorporate regulatory circuits that tie plasmid replication to an environmental signal (perhaps using a conditionally essential replication gene). So far, however, the field has effectively utilized temperature and host-range dependency as the practical means to regulate plasmid presence in LAB.

## 10. Conclusions

Genetic manipulation of lactic acid bacteria has become markedly more feasible. In practice, researchers can now select among complementary DNA-delivery routes (competence-based transformation where available, optimized electroporation in many species, and conjugation or phage-mediated transfer for recalcitrant strains) and pair them with LAB-adapted vectors for expression and clean genome edits. These capabilities have immediate value for food-grade strain improvement and probiotic design.

Priorities ahead are pragmatic: standardize transformation protocols across strains; chart restriction–modification landscapes to guide methylation-aware DNA preparation; expand the set of stable, broad-host-range replicons; and integrate food-grade selection with modern genome-editing platforms. With these advances, tailored yet routine engineering of industrial and probiotic LAB is within reach.

## Figures and Tables

**Figure 1 ijms-26-09146-f001:**
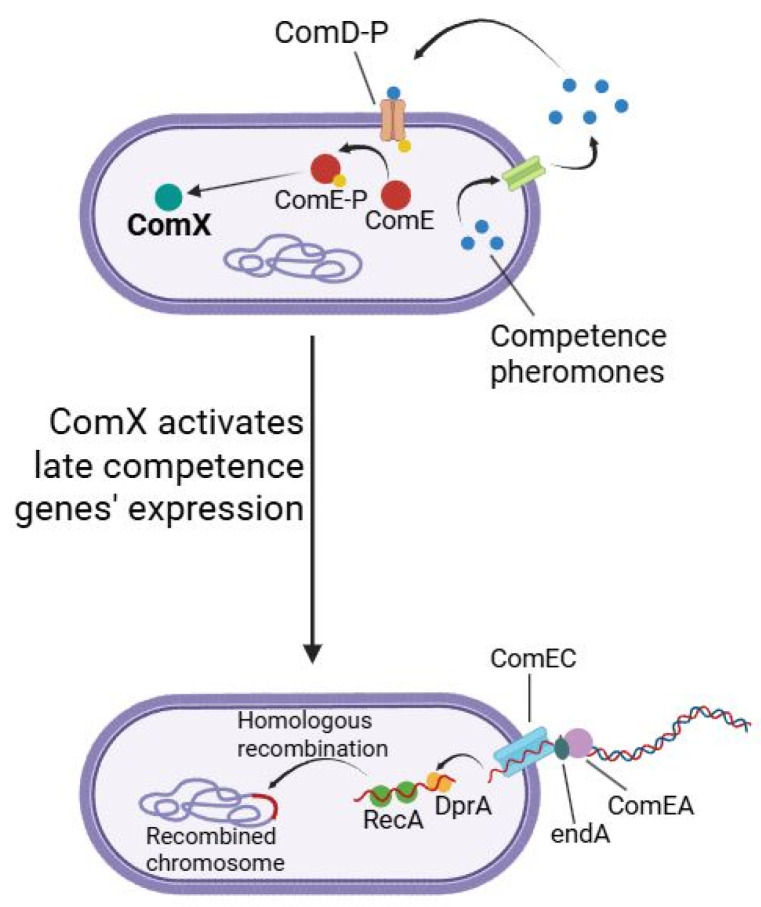
Natural transformation in Gram-positive *Streptococcus pneumoniae*. First, competence pheromones (also known as quorum-sensing peptides) are transferred out of the cell. Then, when their concentration reaches a certain threshold, these pheromones bind to ComD transmembrane complex, triggering its autophosphorylation to ComD-P and a subsequent phosphate transfer to a ComE protein. The phosphorylated ComE-P then activates the transcription of *comABCDE* (creating a positive feedback loop for this whole process) and *comX* genes. After this, ComX activates late competence genes’ expression in the cell, including membrane transporters, nucleases, and recombination apparatus, enabling the competence state of the cell. In this state, the cell can freely take foreign DNA that is coated in RecA and DprA proteins to enable homologous recombination if the DNA passes the defense system of the cell. If no homologous recombination can occur, then the foreign DNA is destroyed.

**Figure 2 ijms-26-09146-f002:**
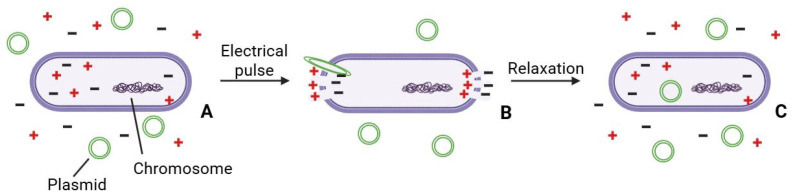
Electroporation in Gram-positive bacteria. (**A**) Plasmids are added to the cell mixture. Then, an (**B**) electrical pulse is applied, due to which pores in the cell walls are formed, and the plasmid can enter the cell. After this, the (**C**) electrical pulse is removed, the pores close, and the cells are put in a relaxed state, with some of them obtaining the plasmid.

**Figure 3 ijms-26-09146-f003:**
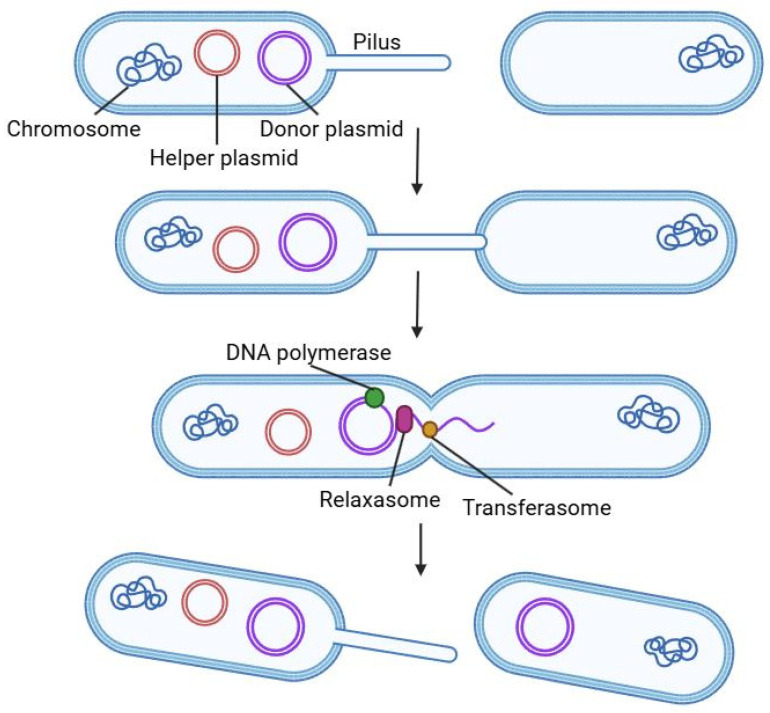
Conjugation in bacteria with the two-plasmid system. After connecting to the recipient cell via pilus conjugation, proteins are expressed from the helper plasmid in the donor cell. After that, donor plasmid replication starts, and this plasmid is immediately transferred with the help of relaxasome and transferasome into the recipient cell.

**Figure 4 ijms-26-09146-f004:**
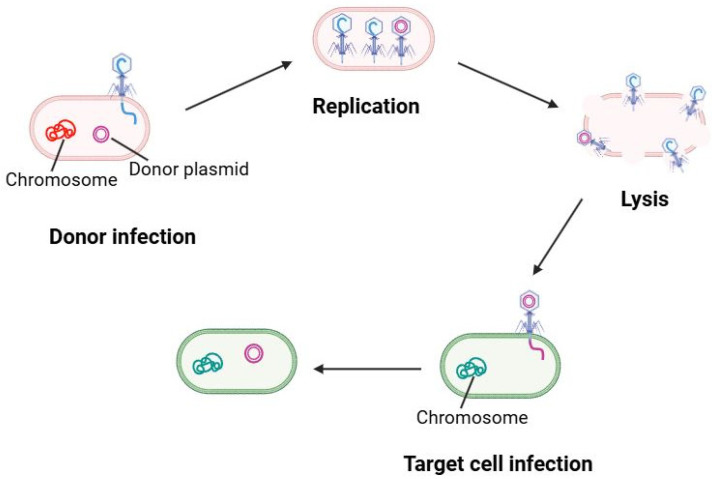
Generalized transduction in bacteria. After infecting the donor cell, the phage begins replication and, during this process, can accidentally take the donor plasmid in its capsid. After lysis and target cell infection, the phages that packaged the donor plasmid instead of their own DNA transfer this plasmid into accepting cells and, being unable to replicate, leave them intact.

**Table 1 ijms-26-09146-t001:** Comparison table of various factors affecting electroporation efficacy in LAB.

Factor	Example Conditions	Reported Effects	Citation(s)
Growth phase	Mid-log or early stationary, strain-dependent	Using exponential or stationary-phase cells is common; should be optimized per strain	[20,21]
Wash/osmoprotection	0.5 M sucrose in washes; ~10% glycerol optional; MgCl_2_ 2–10 mM	Sucrose stabilizes; glycerol can reduce arcing at high densities; Mg^2+^ recommended	[21,23,24,35]
Recovery medium	SR medium vs. GM17/SGM17	~3× more transformants for *L. lactis* BC101 on SR vs. GM17	[34]
Electric pulse tuning	~12.5–15 kV/cm for *L. lactis* BC101/IL1403; 400 Ω > 200 Ω for some *L. casei*	Higher field sometimes needed for later phases; higher resistance improved *L. casei*	[34]
DNA methylation of plasmid	Using Dam^−^/Dcm^−^ (unmethylated) DNA when R–M blocks	Orders-of-magnitude boosts possible	[3]
Glycine (cell-wall weakening)	*L. casei* 0.5–1%; *L. plantarum* ~6–8%	30–100× increase in *L. plantarum*; high efficiencies in many *L. casei* strains	[21,23]
NaCl (osmotic stress)	0.9 M during growth	Up to ~10^5^ CFU/µg in *L. casei*; broadly beneficial	[21,25]
Lysozyme	LM0230: 2000 U/mL, 37 °C, 20 min; BR11: 10 µg/mL, 37 °C, 30 min	30–1000× (LM0230); ~15× (BR11)	[26,28]
Penicillin	10 µg/mL during growth	~480× increase in electrotransformation	[27,28]
DL-threonine	20–40 mM during growth	Enabled *S. cremoris* transformants; ~2× in *P. acidilactici*; 1.6–2.1× in *Clostridium*	[29,30,31]
LiAc/DTT pretreatment	Brief LiAc/DTT exposure before electroporation	Boosted efficiencies in *L. lactis*, *L. plantarum*, *L. buchneri*	[3,32]

**Table 2 ijms-26-09146-t002:** Comparison of DNA introduction methods.

Method	Typical Efficiency & Scope	Major Pros	Major Cons	Best Use Cases	Citations
Natural transformation	Strain-dependent; very high in competent *S. thermophilus*; demonstrated in engineered *L. lactis*	Marker-free edits; clean genome work; regulatory-friendly	Limited host range; careful induction needed	Precise genome edits in competent strains; industry strains where “food-grade” edits matter	[7,13,18]
Electroporation	Commonly 10^4^–10^6^ CFU/µg in optimized strains (*L. lactis*, *L. casei*, *L. plantarum*)	Fast; generalizable; tunable with wall-weakening agents & buffers	Strong strain-to-strain variability; R–M barriers; optimization heavy	Routine plasmid work; recombineering; broadest “first try” method	[21,24,25,34]
Conjugation	Can work when electrotransformation fails; frequencies vary with donor/recipient relatedness	Bypasses some R–M barriers; moves large elements (plasmids/ICEs)	Setup overhead; selectable markers/oriT/relaxase needed	Moving broad traits (lactose use, proteolysis, EPS, bacteriocins) into recalcitrant strains	[36,41,45]
Phage-mediated transduction	Reported for *Lactococcus* (ΦLC3) and *Lactobacillus delbrueckii* (LL-H)	Precise DNA delivery via phage; useful host expansion	Narrow host ranges; requires phage handling	When suitable phage exists and electroporation/conjugation underperform	[54,56,60]
Biolistic particle delivery	Rarely used in LAB; proven in Gram-positives	Host-agnostic physical delivery; can bypass some uptake barriers	Specialized equipment; lower efficiencies and higher mortality vs. electroporation	Last-resort option for hard-to-transform strains	[62,64,66]

**Table 3 ijms-26-09146-t003:** The most known and used LAB vectors and their comparison.

System/Representative Vectors	Primary Hosts	Replicon Family & Class	Induction/Promoter	Selection Marker	Notes	**Citations**
NICE family (pNZ8048/pNZ8148/pNZ8110)	*Lactococcus lactis*	pSH71/pWV01 family, RCR (rolling-circle)—widely used in *lactococci*	Nisin (PnisA/PnisF); NisR/NisK in host	Cm for pNZ8148; Em variants exist	Tight, high dynamic range; secretion variants (Usp45) available	[112,113]
pMG36e	*L. lactis* (often others)	RCR (pWV01-type)	Constitutive P23	Em	Simple, widely used constitutive vector in *L. lactis*	[114]
pTRK/pTRKH2 backbone (pTRK892/pTRK888–890)	Broad *Lactobacillus*	Theta (pAMβ1-type) backbone usually; pTRK888–890 specifically on RCR (pWV01) for broad host	Sugar-inducible (Pfos, Plac, Ptre) or strong constitutive Ppgm	Em	Broad host vectors used in *Lactobacillus*	[115]
pSIP series (pSIP409/pSIP411/pSIP413)	*L. plantarum, L. sakei*	Theta (derived from *L. sakei* sakacin plasmid)	Sakacin P peptide (SppIP) inducible	Em	High-level inducible expression; widely used in *Lactobacillus*	[96,97,98]
pSIP–alr (food-grade variant)	*L. plantarum*	Theta	SppIP	alr complementation (no antibiotic)	Food-grade selection swap for pSIP	[101]
pLEB590	*L. lactis*	RCR (pWV01-type)	Constitutive P45	nisI immunity gene (no antibiotic)	Bacteriocin-immunity-based food-grade selection	[109]
pDBORO	*L. lactis*	Theta	-	oroP orotate utilization (no antibiotic)	Metabolic complementation selection, no antibiotics	[84]
pORI19 + pVE6007 (two-plasmid integration toolkit)	*L. lactis*	pORI19: Ori^+^ RepA^−^ (suicide); pVE6007: pWV01-Ts (temperature-sensitive)	-	Em	Gold-standard for chromosomal insertion/allelic exchange	[116,117]
pG + host9 (single-plasmid Ts integration/counter-selection)	*L. lactis,* many *Lactobacillus*	pWV01-Ts (temperature-sensitive)	-	Em	Widely used Ts “pop-in/pop-out” tool	[118,119]

## Data Availability

Only publicly available datasets were analyzed in this study.

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
