# Peer review of "Advances in Genetic Transformation of Lactic Acid Bacteria: Overcoming Barriers and Enhancing Plasmid Tools"

_ijms, 2025, doi:10.3390/ijms26189146_

Round 1

Reviewer 1 Report

Comments and Suggestions for Authors

The manuscript (ijms-3866775) is well-structured and clearly conveys the scope of the review, covering both challenges and advances in LAB genetic transformation.

Recommendations

  1. The description of methodological advances in abstract (e.g., electroporation efficiencies, competence induction) is detailed and informative. Still, the emphasis supports heavily toward transformation strategies; a brief mention of the broader implications for food or probiotic applications would strengthen the connection to applied relevance.
  2. The final sentences nicely highlight unresolved challenges and emerging approaches. Expanding slightly on how novel genome-editing tools (e.g., CRISPR-based systems) could practically reshape LAB biotechnology would make the abstract more forward-looking.
  3. The introduction section provides excellent coverage of transformation barriers and delivery methods, but the link to practical applications (food fermentation, probiotic functionality, psychobiotics) becomes less prominent as the text progresses. Restating the applied significance at the end would help tie the technical discussion back to the broader motivation.
  4. The indication of genome-editing tools (CRISPR-based systems, recombineering) at the end is useful. However, it feels somewhat unexpected. Expanding this closing statement into a forward-looking perspective would make for a stronger conclusion to the introduction section.
  5. Line 148-151. Repetition occurs in phrases such as “very large DNA inserts (up to 15 kb) have been successfully integrated” followed by “sizes that would be challenging to clone and introduce via plasmid vectors”. This could be simplified into a single concise statement.
  6. The authors should focus on some representation codes such as; Be consistent with gene/protein notation (e.g., DrpA appears here but earlier it was DprA). Standardization avoids confusion.
  7. Use consistent strain designations (e.g., lactis subsp. cremoris KW2 instead of mixing “L. lactis” and “KW2 strain”).
  8. Gene/protein names (e.g., RecA, ComX, nucA) are consistently italicized/capitalized, but some bacterial strain names (e.g., casei, L. lactis) are not always italicized as per convention. The authors recommended to double check through the manuscript.
  9. The authors should pay attention to some casual phrasing, e.g., “big improvement” or bloated and fragile.” For a scientific review, these should be rephrased more formally, e.g., “substantial improvement” or “cells became enlarged and structurally fragile.”
  10. The authors recommended to modify some sentences begin like “It should be noted though that” which could be shortened to “However” for a smoother academic tone.
  11. Ensure consistency in italicization of bacterial species names (Lactobacillus plantarum, L. casei). Some examples are not structured appropriately.
  12. Technical terms like “PFU,” “concatemers,” “cos,” and “pac” are appropriate for an experienced researcher, but you might briefly define them at first use to support interdisciplinary researchers.
  13. Some phrases, such as “blast through” and “workhorse,” are informal and could be replaced with more academic/research tone language.
  14. The authors should consider avoiding informal verbs like “tricks” or “vividly demonstrated” in a scholarly review context.
  15. The authors should check thoroughly and provide the full form of all the abbreviations used at the first mentioned place.

This paper accomplishes the scope of International Journal of Molecular Science (IJMS) In the current form, the manuscript requires major revision. Nevertheless, the revised manuscript can be considered for publication after the inclusion of apposite responses to raised queries.

Author Response

We are grateful to the reviewer for their constructive feedback. We revised the manuscript accordingly.

Comment 1: The description of methodological advances in abstract (e.g., electroporation efficiencies, competence induction) is detailed and informative. Still, the emphasis supports heavily toward transformation strategies; a brief mention of the broader implications for food or probiotic applications would strengthen the connection to applied relevance.

Reply 1: Added two sentences linking transformation advances to starter robustness, probiotic trait engineering, and food-grade editing.

Comment 2: The final sentences nicely highlight unresolved challenges and emerging approaches. Expanding slightly on how novel genome-editing tools (e.g., CRISPR-based systems) could practically reshape LAB biotechnology would make the abstract more forward-looking.

Reply 2: Added a concise look-ahead sentence and cited our dedicated review of LAB genome-editing tools.

Comment 3: The introduction section provides excellent coverage of transformation barriers and delivery methods, but the link to practical applications (food fermentation, probiotic functionality, psychobiotics) becomes less prominent as the text progresses. Restating the applied significance at the end would help tie the technical discussion back to the broader motivation.

Reply 3: Added a short closing paragraph tying methods to fermentation/probiotics and previewing the editing toolkits.

Comment 4: The indication of genome-editing tools (CRISPR-based systems, recombineering) at the end is useful. However, it feels somewhat unexpected. Expanding this closing statement into a forward-looking perspective would make for a stronger conclusion to the introduction section.

Reply 4: Folded a 3-sentence forward-looking mini-perspective into the new closing paragraph (see above).

Comment 5: Line 148-151. Repetition occurs in phrases such as “very large DNA inserts (up to 15 kb) have been successfully integrated” followed by “sizes that would be challenging to clone and introduce via plasmid vectors”. This could be simplified into a single concise statement.

Reply 5: Collapsed to one sentence.

Comment 6: The authors should focus on some representation codes such as; Be consistent with gene/protein notation (e.g., DrpA appears here but earlier it was DprA). Standardization avoids confusion.

Reply 6: Standardized to DprA for the protein.

Comment 7: Use consistent strain designations (e.g., lactis subsp. cremoris KW2 instead of mixing “L. lactis” and “KW2 strain”).

Reply 7: Standardized first use to full binomial + subspecies + strain; abbreviated forms thereafter.

Comment 8: Gene/protein names (e.g., RecA, ComX, nucA) are consistently italicized/capitalized, but some bacterial strain names (e.g., casei, L. lactis) are not always italicized as per convention. The authors recommended to double check through the manuscript.

Reply 8: Audited and corrected throughout: genes nucA, endA, comX/ABCDE (italics); proteins RecA, ComX, SSB (roman/case); all species italicized.

Comment 9: The authors should pay attention to some casual phrasing, e.g., “big improvement” or “bloated and fragile.” For a scientific review, these should be rephrased more formally, e.g., “substantial improvement” or “cells became enlarged and structurally fragile.”

Reply 9: Rephrased to formal scientific style.

Comment 10: The authors recommended to modify some sentences begin like “It should be noted though that” which could be shortened to “However” for a smoother academic tone.

Reply 10: Implemented.

Comment 11: Ensure consistency in italicization of bacterial species names (Lactobacillus plantarum, L. casei). Some examples are not structured appropriately.

Reply 11: Corrected.

Comment 12: Technical terms like “PFU,” “concatemers,” “cos,” and “pac” are appropriate for an experienced researcher, but you might briefly define them at first use to support interdisciplinary researchers.

Reply 12: Added first-use definitions in Transduction (§5) and a consolidated Abbreviations section.

Comment 13: Some phrases, such as “blast through” and “workhorse,” are informal and could be replaced with more academic/research tone language.

Reply 13: Corrected.

Comment 14: The authors should consider avoiding informal verbs like “tricks” or “vividly demonstrated” in a scholarly review context.

Reply 14: Corrected.

Comment 15: The authors should check thoroughly and provide the full form of all the abbreviations used at the first mentioned place.

Reply 15: Implemented and also added an Abbreviations section.

Reviewer 2 Report

Comments and Suggestions for Authors

This comprehensive review article discusses the features of genetic transformation of lactic acid bacteria (LAB), recent advances in this field, and prospects associated with improving these methods in relation to further genome editing of these bacteria. Along with this, the huge diversity of LAB plasmids, both natural and those created on their basis using genetic engineering approaches, is described in detail.

LAB have been used by humans for thousands of years and have been the subject of intense study for over a century. Therefore, a comprehensive review of this topic is indeed a hard task. However, the authors succeeded to focus on the most interesting, relevant and in-demand aspects – DNA delivery and plasmid vectors engineering; in addition, this manuscript contains not only a large amount of information, carefully summarized, but also important generalizations, making the review a valuable tool for any researcher working in this field. The review is well written and easy to read.

 However, minor remarks may be noted as follows.

  1. In the section 2, “Natural transformation and competence”, indeed, the important original articles, mostly, from 90-es to 2000-s are cited. Nevertheless, it is meaningful to mention also the information summarized in the latest review articles, e.g., O’Connell et al, 2022; DOI: 1146/annurev-food-052720-011445 and/or Di Giacomo et al, 2022 doi: 10.1093/femsre/fuac014 or to refer to these articles. Besides, in Line 102, when describing a classical model of the natural transformation, Streptococcus pneumonia, the authors refer to only an article of Blomqvist et al, 2006, https://doi.org/10.1128/AEM.01156-06 (Ref. 7), whereas the reference to the original paper of  Griffith (Griffith F. The significance of pneumococcal types. J Hyg 1928;27:113–59. DOI: 10.1017/s0022172400031879) would have been much more appropriate, especially considering that it was this work that became the starting point for defining the role of DNA as a carrier of genetic information by Avery, McLeod  and McCarty in 1944 - just as a tribute to the great discoveries in molecular biology.
  2. The Section 3, “Electroporation”: In this Section, a very convenient structure of material presentation was chosen, where a number of cell wall weakening chemical agents is described in the context of their effect on cell wall integrity. Meanwhile, physical and biological factors (electric impulse profile, thermal treatment, recovery time, growth phase, etc) are also considered. It is meaningful to add the reference(s) to other review articles summarizing similar data, e.g., Wang et al, 2020, doi 10.1016/j.mimet.2020.105944.
  3. It should be better to add the Section “Abbreviations”, especially considering that one can find some of them in the text before they are explained (for example, Ts).
  4. Despite the text is well-written in general, there are several typos:
  • In Line 69 and below throughout the text it is necessary to indicate the full or abbreviated Latin designations of the strains depending on their appearance in the text (whether mentioned for the first time or not).
  • In Line 115 and below many times throughout the text including the Tables ad Figures, the right designations of genes, proteins, genetic elements should be presented: for example, nucA, endA (for genes); SsbB (for proteins, products of corresponding genes); cos, pac (for phage genetic elements).
  • In Line 557: probably, “… transductants titer…” instead of “… transduction titer…”.
  • In Lines 398, 403: probably, “…lab…” means “…LAB…”.

Author Response

We are grateful to the reviewer for their constructive feedback. We revised the manuscript accordingly.

Comment 1: In the section 2, “Natural transformation and competence”, indeed, the important original articles, mostly, from 90-es to 2000-s are cited. Nevertheless, it is meaningful to mention also the information summarized in the latest review articles, e.g., O’Connell et al, 2022; DOI: 1146/annurev-food-052720-011445 and/or Di Giacomo et al, 2022 doi: 10.1093/femsre/fuac014 or to refer to these articles. Besides, in Line 102, when describing a classical model of the natural transformation, Streptococcus pneumonia, the authors refer to only an article of Blomqvist et al, 2006, https://doi.org/10.1128/AEM.01156-06 (Ref. 7), whereas the reference to the original paper of  Griffith (Griffith F. The significance of pneumococcal types. J Hyg 1928;27:113–59. DOI: 10.1017/s0022172400031879) would have been much more appropriate, especially considering that it was this work that became the starting point for defining the role of DNA as a carrier of genetic information by Avery, McLeod  and McCarty in 1944 - just as a tribute to the great discoveries in molecular biology.

Reply 1: We are grateful to the reviewer for pointing out this additional historical context. Added O’Connell et al., 2022, Di Giacomo et al., 2022, and the classic Griffith, 1928 historical reference with one-sentence context.

Comment 2: The Section 3, “Electroporation”: In this Section, a very convenient structure of material presentation was chosen, where a number of cell wall weakening chemical agents is described in the context of their effect on cell wall integrity. Meanwhile, physical and biological factors (electric impulse profile, thermal treatment, recovery time, growth phase, etc) are also considered. It is meaningful to add the reference(s) to other review articles summarizing similar data, e.g., Wang et al, 2020, doi 10.1016/j.mimet.2020.105944.

Reply 2: Added as an umbrella reference in §3, first paragraph.

Comment 3: It should be better to add the Section “Abbreviations”, especially considering that one can find some of them in the text before they are explained (for example, Ts).

Reply 3: New Abbreviations section added.

Comment 4: In Line 69 and below throughout the text it is necessary to indicate the full or abbreviated Latin designations of the strains depending on their appearance in the text (whether mentioned for the first time or not).

Reply 4: Implemented.

Comment 5: In Line 115 and below many times throughout the text including the Tables ad Figures, the right designations of genes, proteins, genetic elements should be presented: for example, nucAendA (for genes); SsbB (for proteins, products of corresponding genes); cospac (for phage genetic elements).

Reply 5: Implemented.

Comment 6: In Line 557: probably, “… transductants titer…” instead of “… transduction titer…”.

Reply 6: Revised to “titer of transducing particles” to remove ambiguity.

Comment 7: In Lines 398, 403: probably, “…lab…” means “…LAB…”.

Reply 7: Corrected.

Round 2

Reviewer 1 Report

Comments and Suggestions for Authors

The manuscript addresses an important and timely topic on the genetic transformation of lactic acid bacteria (LAB), highlighting both current barriers and recent advances in plasmid-based tools. The subject is highly relevant for food biotechnology, probiotics, and synthetic biology.

Mentioning transformation efficiency numbers (10⁴–10⁶) or specific gene names (ComX, synthetic peptides) may be too detailed for the abstract, better kept inside the main text.

The last sentence “see our recent review [1]” is usually discouraged, abstracts should be self-contained.

The conclusion section, it’s still quite long, almost like a mini-review inside the conclusion. A bit of summarizing would improve readability without losing detail. For example, combining some sentences on electroporation and plasmid vectors.

Some ideas; e.g., barriers-solutions, electroporation limitations, plasmid host-specific factors are mentioned twice in slightly different forms. This could be shortened.

Language and grammar are generally fine, but minor polishing would improve readability and flow. Shortening too long sentences will help clarity.

Author Response

We are grateful to the reviewer for their constructive feedback. We revised the manuscript accordingly.

Comment 1: Mentioning transformation efficiency numbers (10⁴–10⁶) or specific gene names (ComX, synthetic peptides) may be too detailed for the abstract, better kept inside the main text.

Reply 1: We agree. We simplified the Abstract to keep it self-contained and high-level, removing numerical transformation efficiencies and specific gene/tool names. Details remain in the body and tables.

Comment 2: The last sentence “see our recent review [1]” is usually discouraged, abstracts should be self-contained.

Reply 2: Done. We removed that sentence from the Abstract and instead keep the cross-reference in the Introduction where appropriate.

Comment 3: The conclusion section, it’s still quite long, almost like a mini-review inside the conclusion. A bit of summarizing would improve readability without losing detail. For example, combining some sentences on electroporation and plasmid vectors.

Reply 3: We shortened and consolidated the Conclusions to two compact paragraphs that avoid repeating earlier sections while keeping the key take-homes and forward look.

Comment 4: Some ideas; e.g., barriers-solutions, electroporation limitations, plasmid host-specific factors are mentioned twice in slightly different forms. This could be shortened.

Reply 4: Done. The previous conclusions repeated barriers, electroporation, and plasmid points already covered in the main text. We shortened the section as mentioned in the previous point.

Comment 5: Language and grammar are generally fine, but minor polishing would improve readability and flow. Shortening too long sentences will help clarity.

Reply 5: We performed line-edits to improve readability (split long sentences; remove stacked clauses; tighten transitions).